# Stratified Diabetes Mellitus Prevalence for the Northwestern Nigerian States, a Data Mining Approach

**DOI:** 10.3390/ijerph16214089

**Published:** 2019-10-24

**Authors:** Musa Uba Muhammad, Ren Jiadong, Noman Sohail Muhammad, Bilal Nawaz

**Affiliations:** 1Department of Information sciences and Technology, Yanshan University, Qinhuangdao 066000, China; 2State Key Laboratory of Metastable Materials Science and Technology, Yanshan University, Qinhuangdao 066004, China

**Keywords:** age, gender, diabetes mellitus, Nigeria, classification, prevalence, diagnosed, K-means, real-life data

## Abstract

An accurate classification for diabetes mellitus (DBM) allows for the adequate treatment and handling of its menace, particularly in developing countries like Nigeria. This study proposes data mining techniques for the classification and identification of the prevalence of diagnosed diabetes cases, stratified by age, gender, diabetic conditions and residential area in the northwestern states of Nigeria, based on the real-life data derived from government-owned hospitals in the region. A K-mean assessment was used to cluster the instances, after 12 iterations the instances classified out of 3022: 2662 (88.09%) non-insulin dependent (NID), 176 (5.82%) insulin-dependent (IND) and 184 (6.09%) gestational diabetes (GTD). The total number of diagnosed diabetes cases was 3022: 1380 males (45.66%) and 1642 females (54.33%). The higher prevalence was found to be in females compared to males, and in cities and towns, rather than in villages (36.5%, 34.2%, and 29.3%, respectively). The highest prevalence among the age groups was in the age group 50–69 years, which constituted 43.9% of the total diagnosed cases. Furthermore, the NID condition had the highest prevalence of cases (88.09%). These were the first findings of the stratified prevalence in the region, and the figures have been of utmost significance to the healthcare authorities, policymakers, clinicians, and non-governmental organizations for the proper planning and management of diabetes mellitus.

## 1. Introduction

The recent advances in biotechnology and health sciences have led to a significant production of data such as clinical information generated from massive electronic health records. Machine learning methods have been successfully applied several times in medical domains, for example, in the diagnosis of diabetes aspects and epidemiological studies [1]. Epidemiological studies are based on a data mining approach, which constitutes machine learning 60%, statistics 35% and probability 5%, according to various studies [2].

Nigeria is a country located in the western part of Africa, and with a population of approximately 200 million it is the seventh largest population in the world [3]. The data from the United Nations reported that Nigeria’s population will reach 411 million by 2050 [4]. The northwestern region of Nigeria is the most densely populated area among the six geopolitical zones in the country, with an estimated population of 45 million people [5].

According to the National Population Commission of Nigeria (NPC), approximately 64% of the population live in a rural area, and only 36% live in an urban area [6]. The United Nations (UN) estimate for 2010 was that 49.8% live in a metropolitan area, while 50.2% live in a rural area [6]. As stated by the World Health Organization (WHO), Nigeria has the highest number of people living with diabetes in Africa. The prevalence in Nigeria varies from 0.65% in rural areas to 11% in urban areas [7]. 

Diabetes mellitus (DBM) is a chronic disease that arises either when the pancreas does not produce enough insulin, or when the body cannot efficiently use the insulin it produces. Insulin is a hormone that controls blood sugar. Hyperglycemia or higher blood sugar is a common effect of uncontrolled diabetes and, over time, this leads to severe damage to systems throughout the entire human body [1]. The disease assumes pestilence proportion worldwide, by affecting both developed and underdeveloped countries [5]. The recent prevalence figure published by the International Diabetes Federation is 425 million people living with diabetes mellitus comprehensively, with virtually 50% as non-diagnosed cases [1]. 

Several factors are cited for such high prevalence in the country, including urbanization and change of lifestyle, with an increase in caloric consumption among the populace [8]. Malnutrition and long-time malnutrition related to the socioeconomic setting of poverty, can result in fibro calculus-pancreatic diabetes (FPD) or protein-deficient pancreatic diabetes (PDPD) [9]. Poor compliance to treatment measures by the diabetic patients, and the economic expenses for the management of diabetes complications are far beyond that of the average individual [10]. 

There is also an increasing burden from the complications of diabetes alongside the ever-increasing prevalence of the disease. We now see high rates of diabetes-related amputations, cerebrovascular disease, heart-related problems, and kidney disease in populations that were not previously known for these challenging health problems [11]. 

In recent decades in Nigeria, diabetes has been connected with the renaissance of tuberculosis and the increasing prevalence of end-stage kidney disease. On medical wards, patients with diabetes have the most prolonged hospital stay with relatively expensive medical bills and several additional complications, such as stroke, heart failure, and lower extremity amputation (LEA) from foot gangrene. Diabetes cases have been observed as the most substantial proportion of all admissions in Nigerian medical (nonsurgical) wards and the disease is one of the leading factors causing operative premature births, obstetric delivery, and neonatal mortality [12].

In recent times, related research was conducted—a cross-sectional survey to assesses the prevalence of type 2 diabetes mellitus on socio-demographic and behavioural characteristics of the study population, STEPS survey, Punjab, India, stratified by sex, gender, residential status and social groups (2014–2015) [13]. A very similar study was carried out in Northeastern China to determine the prevalence of pre-diabetes diabetes and the associated risk factors, in which a multistage stratified cluster sampling method was applied to select adults from the Jilin Province [14]. 

An investigation into the potential association between socioeconomic status (SES) and glycemic control in type 2 diabetes was held, using a clinical sample of White and Black patients with type 2 diabetes mellitus in the United States. The study was mainly aimed at the differential role of SES factors upon glycemic control based on race, gender, and their intersection [15]. An initial stratified prevalence of diabetes data for the Republic of Macedonia was derived from the National eHealth system, stratified by age, gender and place of living [16]. 

Recently, several studies on diabetes in Northwestern Nigeria were accompanied with a plan to predict diabetes mellitus, the models utilized information regarding other chronic diseases using datamining-based approaches [5]. Another related study was carried out to determine the prevalence of diabetes mellitus, and its correlation to the suburban population of Sokoto, Northwest Nigeria [8]. In 2016, a ten-years retrospective study was designed to describe the clinical presentation and management outcomes of childhood diabetes mellitus, as seen in Usmanu Danfodiyo University Teaching Hospital, Sokoto, Northwestern Nigeria [9]. 

However, despite the growing prevalence of the diseases particularly in this region and considering the population scope, to the best of our knowledge, there was very little research and awareness of the issue. Due to this, we planned to accomplish this research in order to determine the prevalence of diagnosed diabetes cases stratified by age, gender, residential area and diabetic condition, a data-mining-based approach: a case study of the northwestern part of Nigeria. 

## 2. Materials and Methods 

### 2.1. Ethical Approval

The research was approved by the ethical committee from the four northwestern states, where the research was conducted. The approval was granted from their relevant ministries of health and grant codes are as follows: Jigawa State—FMC/BKD/CLN/HREC/138; Kano State—MOH/ADM/744/VOL.I/558; Katsina State—MOH/ADM/SUB/1152/214; Zamfara State—HSMB/SUB/540/VOL.I. Additionally, all participants provided written informed consent after all of the procedures were explained to them.

### 2.2. Research Design

The study is a cross-sectional design [17] concerning people living with diabetes mellitus, from four out of the seven northwestern states of Nigeria. The states were Jigawa, Kano, Katsina and the Zamfara States, a total of 3022 diabetes patients were evaluated from 2017–2019.

### 2.3. Inclusion Criteria

The study only considered some specific government hospitals/health centres from the states, between April 2017 and June 2019. 

### 2.4. Exclusion Criteria

Private hospitals/health centres and diagnosed cases performed before April 2017 and after June 2019 were excluded.

### 2.5. Data Collection

Real-life data were collected from both primary and secondary sources, from the selected hospitals/health centres in the states. The authors distributed questionnaires to the diabetic mellitus patients in all centres. We also held a verbal interview for the inclusion of those who could not write, with the help of the hospital staff. Some parts of the hospital’s records which were related to disease symptoms and their complications were also used. The total number of 3022 diabetes patients were evaluated.

The datasets collected comprised of 3044 observations and some attributes. This study only considered the attributes AGE, GEN, DCD and RDS, and these are described below:
*AGE*—Patient’s age (numeric).*GEN*—Patient’s gender (categorical: M—male and F—female).*DCD*—Patient’s diabetes condition (categorical: IND—insulin dependent, NID—non insulin dependent and GTD—gestational).*RSB*—Patient’s residential Suburb (categorical: VL—village, TW—town and CY—city).

The datasets were derived from the exercise, we considered the patient’s age, gender, diabetic condition and residing place. The diagnosed diabetes cases were stratified in the following age groups: below 20 years, 20–39 years, 40–59 years, 60–79 years, and 80 years and above. Following this, we sub-stratified by gender, diabetic condition and location of residence.

The data collection flow for the diabetic patients from the four northwestern states of Nigeria, namely Jigawa (22.53%), Kano (34.31%), Katsina (26.24%) and Zamfara (18.23%), is presented in Figure 1 below.

### 2.6. Classification Accuracy

Classification accuracy is one of the most common tasks in machine learning [18], and one of the most problematic aspects includes unknown instances [19] in one of the pre-offered categories/classes [20]. The critical observation of classification is that target functions are discrete [21]. In general, the class label cannot be meaningfully assigned a numerical or any other value [19], meaning that the class attribute whose value should be determined is a categorical attribute [22].

The classification of an object is based on finding resemblances with a set of objects that are members of different classes, with the similarity of the two objects determined by analyzing their characteristics. In classification, every object is classified into one of the classes with high accuracy by the machine learning algorithm, J48, with an accuracy of 99.28% [23]. The task is to make a model by which will be performed a classification of new objects [24], based on the characteristics of objects whose classification is known in advance.

The classification assessment of the J48 classifier with confusion matrix accuracy is explained by Equations (1) and (2) and is shown below: (1)$$arg(min)\overset{k}{\underset{j=1}{\sum }}\frac{1}{2\left|s\right|}\underset{x,y\in s}{\sum }\|x-y\|$$
(2)Accuracy=(TP+TN)/(TP+TN+FP+FN)
Correctly classified instance: 3022 = 99.28%

Incorrectly classified instance: 22 = 0.72%

Kappa statistics: 0.9823

Mean absolute error: 0.0072

Relative absolute error: 0.0166

Root relative square error: 0.1904


**Confusion Matrix accuracy**


**Table d35e505:** 

a	b	Classified as:
512	2	a = correctly
20	2510	b = incorrectly

### 2.7. K-Means Clustering

Clustering is an efficient data mining technique to classify the intrinsic structure that lies behind the given data objects (points) [25]. It refers to the process of organizing the given objects into homogeneous classes called clusters, whose members are similar to each other. The K-means algorithm is a popular method among all the partitioning clustering techniques. 

The K-means algorithm [26] is explained in the following steps:

Step 1: Select L initial cluster, centres $c_{1},c_{2},\ldots ,c_{l}$ randomly from n point
(3)$$\left\{x_{1},x_{2},\ldots ,x_{n}\right\}L\leq n$$

Step 2: Assign each point $x_{i}:i=1,\;2,\ldots ,n$ to the cluster $C_{j}$ corresponding to the cluster centre $c_{j}$ for j=1,2,…,l
iff
(4)$$\left\|x_{i}-c_{j}\|\leq \|x_{i}-c_{p}\right\|,\;p=1,2,\ldots ,l\;\mathrm{and}\;j\neq p$$

Step 3: Compute new clusters $c_{1}{}^{\prime },c_{2}{}^{\prime },\ldots ,c_{i}{}^{\prime }$ as
(5)$$c_{i}{}^{\prime }=\frac{1}{n_{i}}\sum_{x_{j}\in C_{i}}x_{j}\;\mathrm{for}\;i=1,2,\ldots ,l.$$
where $n_{i}$ is the number of data points belonging to the cluster $C_{i}$.

Step 4: If $c_{i}{}^{\prime }=c_{i},\forall$
i=1, 2,…,l then terminate. Otherwise, continue from (4).

### 2.8. Study Analytical Platform

Initially, the machine learning method was utilized for data extraction and integration. Additionally, this was used for grouping the instances by age, gender, and place of residence. R-programming software and Waikato Environment for Knowledge Analysis were utilized for the classification and clustering assessment of diabetes mellitus conditions with an efficient K-means clustering algorithm used to assign the classes [26].

Three methods were used to collect the data. These were a questionnaire, verbal interview and information from the hospital’s record department. Later, all data were combined in an excel sheet without exception, and the total number of patients was 3044.

All patients were carefully examined and classified accordingly, for the initial screening by proper prognosis assessment [27]. After data preprocessing, the final dataset included 3022 patient records with 45.7% males and 54.3% females, and four attributes namely age, gender, residential area, and diabetic condition, while NID, IND and GTD were the classes of diabetes. The population sampling included patients with diabetes mellitus status type 1 (insulin-dependent), type 2 (non-insulin dependent), and gestational diabetes. All four attributes were classified according to the research motive, and the results are presented in Section 3.

## 3. Results

The following are the results from the population sample of Northwestern Nigerian diabetic patients. The population comprised of a total of 3022 patients diagnosed with different conditions who were classified and the results are presented in both tables and figures.

The data collection flow for diabetic patients from the four northwestern states of Nigeria is presented in Figure 1. The analytical study platform is shown in Figure 2, the entire process can be described by three stages. The initial stage, objective disease screening, consists of data collection, extraction and integration. The second phase is a data mining platform, and in the final stage, the objectives were achieved through evaluation, assessment and clinical implications.

Table 1 presents the diabetes conditions and the number of patients classified in clusters assessment, and their various percentage ratios.

Age groups classified the patients, age < 20 patients, ages 20–39 equaled to 676 patients, ages 40–59 equaled 1328 patients, ages 69–79 equaled 934 patients, and ages 80 and above equaled 64 patients. The K-means cluster assessment for the patient’s diabetic conditions is presented in Figure 3 below. 

The K-means clustering algorithm was accomplished after twelve iterations, three non-overlapping subgroups were formed each with centroid lies in the centre of the clusters. As observed, the groups represent three diabetic conditions, and these are non-insulin dependent (NID), insulin-dependent (IND) and gestational diabetes (GTD). The patient’s diabetic condition and age distribution flows are presented in Figure 4 and Figure 5. 

Table 2 presents the cases stratified according to; the age, gender, residential area, and diabetic condition. The total number of diagnosed cases was 3022 individuals, of which 1380 were males (45.7%) and 1642 were females (54.3%). The mean age for all diagnosed cases was 50.9 ± 10.5 years.

The highest number of diagnosed diabetes cases was found in the age group 40–59 years, in total, this was 1328 patients. This was followed by the age group 60–79 years with 934 cases, then the age group 20–39 years with 676 cases, the age group 80 years and above with 64 instances, and finally, the age group below 20 years with only 20 cases. 

The highest occurrence of diagnosed cases found was recorded in the age group 40–59 years, in males and females (18.96% and 24.98%, respectively), followed by the age group 60–79 years (18.20% and 12.71% in males and females, respectively), age group 20–39 years (6.95% and 15.4% in males and females, respectively), age group 80 years and above (1.2% and 0.96% in males and females, respectively) and the age group below 20 years (0.13% and 0.13% in males and females, respectively).

Table 3 presents the summary of diabetes cases by age, gender, residential area and complications. The total diagnosed cases stratified according to the residential areas are found to be higher in cities with 1102 cases (36.47%), followed by towns with 1034 cases (34.22%), and finally, villages with 886 cases (29.32%). Additionally, the most commonly diagnosed diabetic condition happens to be the non-insulin dependent diabetic condition, with a total number of 2662 cases (88.09%). Insulin-dependent and gestational diabetes were found to be 174 cases (5.82%) and 186 cases (6.09%), respectively.

## 4. Discussion

The study employed a data mining platform in the classification processes of the collected samples of diabetes mellitus patients from the northwestern states of Nigeria. In this paper, the J48 machine learning and K-means clustering algorithms were performed on the dataset to classify the instances according to the desired research motive. The cases are organized and presented in Table 1 and Figure 3 and Figure 5.

The J48 machine learning classifier classified 3022 instances correctly with an accuracy of about 99.28%, while 22 instances were removed as outliers during the experiment. The purpose of using R was to standardize the datasets before executing the experiment and was used to classify the instances. The result from R was used on WEKA for clustering the instances.

The study further explained the prevalence of diagnosed diabetes cases stratified by age, gender, residential area and diabetic conditions in Table 2 and Table 3. Diabetes is an enormous healthcare and socio-economic burden of the region and Nigeria as a whole. The stratified prevalence figures have been of utmost importance for all stakeholders, e.g., healthcare authorities, policymakers, clinicians, and non-governmental organizations. Since there are no reliable data sources, only these were available [28]. 

According to the International Diabetes Federation (IDF), the highest prevalence of diabetes in Nigeria, Africa, is found between ages of 55 and 64 years. While the most prevalent diabetes condition found is type 2, accounting for 90–95% of the total diagnosed cases [28]. This validates our study findings of the highest prevalence occurring in the age group 40–59 years, with a total of 1328 out of 3022 diagnosed cases (43.94%) including both genders. Likewise, the diabetic condition was non-insulin dependent (type 2 diabetes) and this was found to be predominantly the situation, with a total of 2662 cases (88.09%).

The urbanization process has been playing a significant role in most of the northwestern Nigerian cities, habitants of rural areas over a period of time will usually migrate to urban areas for one reason or another. As a result, a high prevalence of diagnosed cases significantly resides in the villages, towns and cities, totaling 886 (29.32%), 1034 (34.22%), and 1102 (36.47%), respectively. Also, there was a higher number of cases recorded in females compared to males; 1642 cases (54.3%) vs. 1380 cases (45.7%), respectively. This could be explained by the fact that females conventionally live at home to maintain the household and are physically less active, while males are physically more active and engaged in office work, businesses and other activities like farming and rearing animals.

The main strength of this study is the use of a real-life dataset, which was collected from four northwestern states of Nigeria. Moreover, a limitation is the processing time in running the experiment, which depends on the type of datasets used and the number of patients included. 

## 5. Conclusions

This study implements the J48 machine learning rule classifiers and K-means clusters algorithms on a data mining platform to classify potential diabetes conditions in the initial disease screening of a patient, for the possible evaluation and assessment of the clinical implications.

The results are the first findings on the prevalence of diabetes in the northwestern states of Nigeria, including diagnosed cases and stratified by age, gender, residential area, and diabetic conditions. It has been observed that the higher prevalence of cases was found in the age group 40–59 years, predominantly the non-insulin dependent form of the condition otherwise known as type 2 diabetes. Additionally, the recorded instances happened more frequently in cities and among females.

Furthermore, the stratified prevalence figures have been of the utmost significance to healthcare authorities, policymakers, clinicians, and non-governmental organizations for the proper planning and management of diabetes mellitus. Therefore, such a study is found to be remarkably crucial for the states, region and the nation as a whole.

## Figures and Tables

**Figure 1 ijerph-16-04089-f001:**
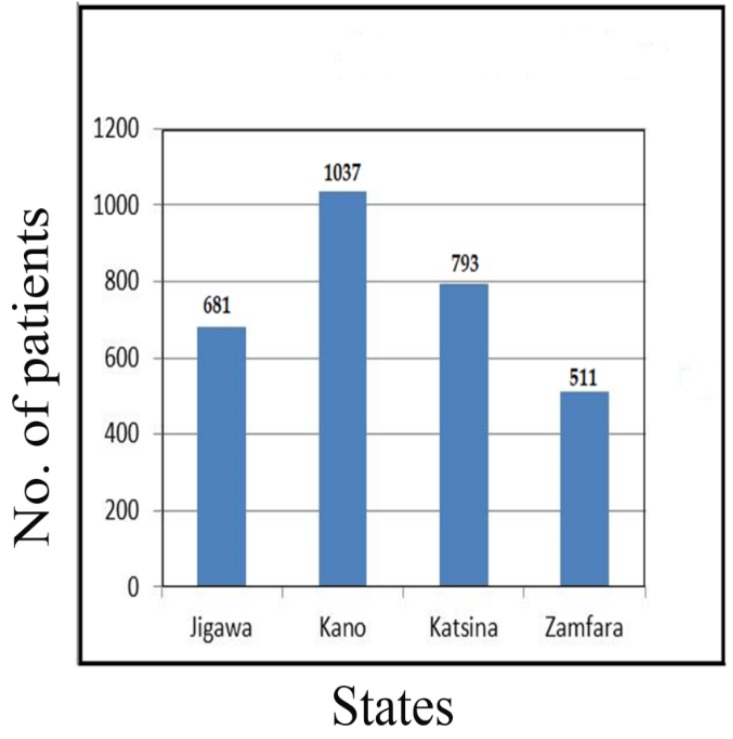
Data collection flow.

**Figure 2 ijerph-16-04089-f002:**
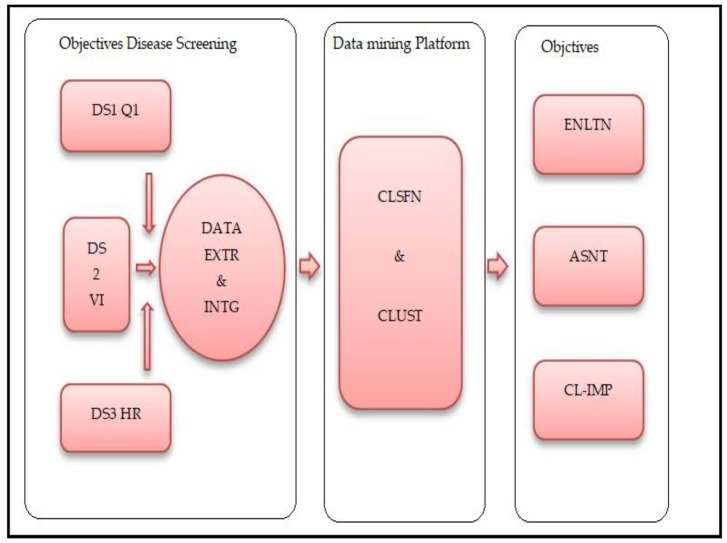
Study analytical platform. Legend: DS1QN: Data Source One Questionnaire, DS2VI: Data Source Two Verbal Interview, DS3HR: Data Source Three Hospital Records, EXTR: Extraction, INTG: Integration, STATS: Statistics, R: R-Programming Software, WEKA: Waikato Environment, CLUST: Clustering, CLSFN: Classification, EVLTN: Evaluation, ASMT: Assessment, CL-IMP: Clinical-Implications.

**Figure 3 ijerph-16-04089-f003:**
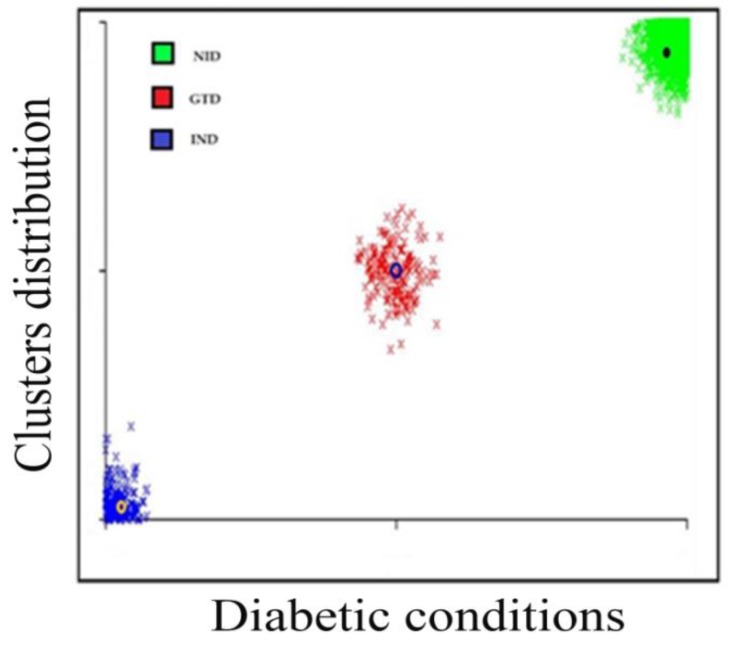
K-means cluster assessments for the patient’s diabetic conditions.

**Figure 4 ijerph-16-04089-f004:**
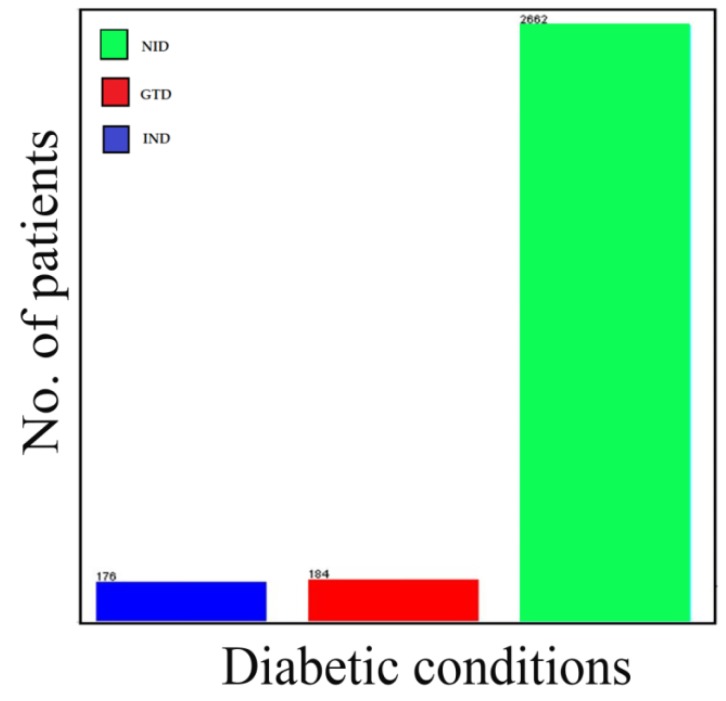
Bar flow for the patient’s diabetic condition.

**Figure 5 ijerph-16-04089-f005:**
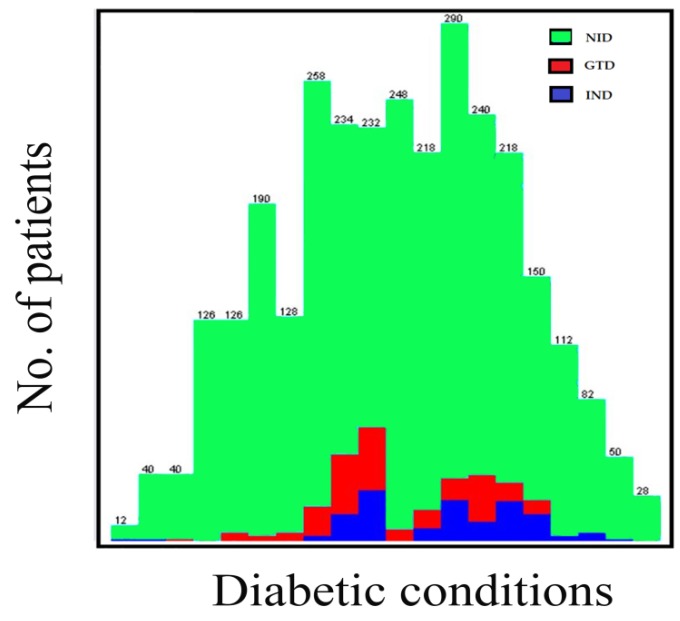
Patients age distribution based on diabetic conditions.

**Table 1 ijerph-16-04089-t001:** Diabetes conditions and the number of patients classified in the assessment.

Diabetic Condition	Patients *N* = 3022	Age	Ratio		Cluster by Diabetic Conditions		
NID	2662	12 ≤ x ≤ 85	88.09%	“0” missing-value	NID	IND	GTD
IND	176		5.82%		2662	176	184
GTD	184		6.09%				

*N* = number of diabetes patients, x = patients age, NID = non-insulin dependent, IND = insulin dependent, GTD = gestational diabetes.

**Table 2 ijerph-16-04089-t002:** Diagnosed diabetes cases stratified by age, gender, residential area and diabetic conditions.

Age (Years)	Residential Area	Gender				Diabetic Condition					
		M	%	F	%	NID	%	IND	%	GTD	%
<20	City	8	0.26	4	0.13	13	0.43	3	0.10	0	0
	Town	0	0	0	0	0	0	0	0	0	0
	Village	4	0.13	4	0.13	3	0.10	1	0.03	0	0
	Total	12	0.39	8	0.26	16	0.53	4	0.13	0	0
20–39	City	101	3.34	205	6.78	295	9.76	1	0.03	10	0.33
	Town	73	2.42	155	5.13	220	7.28	0	0	8	0.26
	Village	36	1.19	106	3.51	128	4.24	2	0.07	12	0.39
	Total	210	6.95	466	15.42	643	21.28	3	0.10	30	0.99
40–59	City	185	6.12	339	11.22	465	15.39	20	0.66	39	1.29
	Town	188	6.22	229	7.58	358	11.85	40	1.32	19	0.62
	Village	200	6.62	187	6.19	327	10.82	19	0.62	41	1.36
	Total	573	18.96	755	24.98	1150	38.05	79	2.61	99	3.28
60–79	City	165	5.46	95	3.14	257	8.50	1	0.03	2	0.07
	Town	195	6.45	156	5.16	287	9.50	45	1.49	19	0.62
	Village	190	6.29	133	4.40	249	8.24	44	1.46	30	0.99
	Total	550	18.20	384	12.71	793	26.24	90	2.98	51	1.69
80+	City	0	0	0	0	0	0	0	0	0	0
	Town	18	0.60	20	0.66	38	1.26	0	0	0	0
	Village	17	0.56	9	0.30	22	0.73	4	0.13	0	0
	Total	35	1.2	29	0.96	60	1.99	4	0.13	0	0

**Table 3 ijerph-16-04089-t003:** The summary of the diabetes cases by age, gender, residing place and diabetic conditions.

Age (Year)	Residing Place	Both Gender (M and F)		Diabetic Condition					
		Total	%	NID	%	IND	%	GTD	%
All	City	1102	36.47	1030	34.08	23	0.76	53	1.75
	Town	1034	34.22	903	29.88	84	2.78	47	1.56
	Village	886	29.32	729	24.12	69	2.28	84	2.78
	Total	3022	100.01	2662	88.09	176	5.82	184	6.09

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
