# Peer review of "Stratified Diabetes Mellitus Prevalence for the Northwestern Nigerian States, a Data Mining Approach"

_ijerph, 2019, doi:10.3390/ijerph16214089_

Round 1
Reviewer 1 Report
The authors have compiled an informative paper to determine the prevalence of diagnosed diabetes cases stratified by age, gender, residential places and diabetic conditions in a case study of the northwestern part of Nigeria. They have used a data mining based approach and have discussed the significance to the healthcare authorities, policymakers, clinicians, and non-governmental organizations for proper planning and management of diabetes mellitus.
Work presented here is interesting but needs more refinement in presentation as well as in discussion.
Major Comments
Introduction needs to be overall edited to read more refined and crisp. Introduction has no mention of the approach used (machine learning). To analyze 3022 patient samples, is it really important to use this approach over the conventional statistical tools used in epidemiological studies? A rationale be added to the introduction to substantiate this. Method section:3a. Classification accuracy
Please explain which method was used. For example, SVM, expected maximization, or neural network (Line 131-135).
3b. What is the accuracy metric used (Line 138-140)
3c. Was any error function used?
3d Lines 160-169, was any Outlier removed? Please describe why R and WECA was used together?
3e. Any normalization was carried out? What do u mean by preprocessing data (line 165)
3f. Mentions about 4 attributes – can see only 3. Please explain the 4 (patients with diabetes mellitus status Type 1 (insulin-dependent), Type 2 (non-insulin dependent), and gestational diabetes ) line 168
All figures needs the axis to be labelled and appropriately use the colors.
4a. Fig 1 needs axis labelling and softer colors for bars preferable
4b. Fig 2 The central panel has STATS, R and WECA mentioned, needs to be eliminated. The central panel should have only cluster and classification which are the data mining platforms.
4c. Fig 3 needs axis labelling. Please define what X and Y axis means.
4d. Fig 4 needs axis labelling. The groups be color coded and followed consistently in all figures. NID= Green, GTD= Red, IND=Blue
4e. Fig 5 needs axis labelling.
Results section:
All the figures needs to be explained well in the result section in a story telling fashion.
5a. Fig 1 and Fig 2 explanation missing
5b. Avoid saying Fig 4 below represents …Line 189. Instead, the result section needs to be re-written cohesively.
5c. Avoid using bullet points (Line 177-181), instead explain it into a paragraph.
Will separating the genders for each figure add new information? Especially considering GDM for female population only. Have you tried to represent all the figures in male and female separate information? The discussion lacks proper drawing of conclusion from the sophisticated analysis performed. Focus on the scientific findings and discuss the points. The article needs to be edited by English native professional editor and taken care for the minor type set errors.
Reviewer 2 Report
"The remaining article is structured as follows: Section 2 presents the material and methodology after Section 3 reviews the results, Section 4 discusses the results and Section 5 concludes the findings".
I think that is not necessary the aclaration.
"The study only considered some specific government hospitals/health centres from the states, between April’ 2017 to June’ 2019"
Can you specify more?
I understand that data is obtained from surveys. The author distributes questionnaires to the diabetic mellitus patients in all centers. 3022 responses were received, but it is not clear to me the total number of diabetic patients to whom the survey was sent. I do not understand the percentage of responses.
The representativeness of the results of this study is not clear. An algorithm would be necessary where the total population appears, the number of diabetic patients registered and the number of patients who answered the survey.
Figure 1 could be replaced by a map of the country with the regions evaluated to see its distribution in the country. Or, it would be interesting to present in a table the socio-demographic characteristics of the areas of the country evaluated against the areas that have not been studied.
Figure 3 should explain what NID, GTD, IND means.
Round 2
Reviewer 1 Report
1. J48 is the name of a decision tree-based model. So, it is better to describe it a little more not just the name of it. The clustering model is explained well, please fix the classification model also.
2. There are some measuring metrics like accuracy, precision, recall, etc. and all of them have formulation. So, it is better to show up your metric (accuracy) with the formulation.
3. About the error function (like root mean square error), please give formulation or some description; each classification method used error function to classify the data based on this error and then get good accuracy in the results.
4. Please put some of your answers in the paper for readers.
No outlier removed during the experiment. The purpose of using R was to standardize the datasets before execution the experiment, and was used to classify the instance. The result from R was used on WECA for clustering the instance.
5. In Table 2, please clarify the main rows (based on age); different colors (gray and white) helps for reader to find the data easily. Table 1 and Table 3 are OK, as they are small.
6. Do a thorough spell check and English correction. All figures the spelling for Diabetic needs to be corrected, Daibetic is wrong.
Reviewer 2 Report
With the corrections made, I consider that it is suitable for publication. In Figures 3, 4 and 5 there is an MISTAKE in the word "DAIBEIC" WHICH "DIABETIC" SHOULDAuthor Response
Please see attachment.
